# Metabolomic Biomarkers in Bovine Embryo Culture Media and Their Relationship with the Developmental Potential of In Vitro-Produced Embryos

**DOI:** 10.3390/ijms26052362

**Published:** 2025-03-06

**Authors:** Elina Tsopp, Kalle Kilk, Andres Gambini, Ants Kavak, Esta Nahkur, Anni Viljaste-Seera, Haldja Viinalass, Ülle Jaakma

**Affiliations:** 1Chair of Animal Breeding and Biotechnology, Institute of Veterinary Medicine and Animal Sciences, Estonian University of Life Sciences, 51006 Tartu, Estonia; elina.tsopp@emu.ee (E.T.); anni.viljaste-seera@emu.ee (A.V.-S.); haldja.viinalass@emu.ee (H.V.); 2Department of Biochemistry, Institute of Biomedicine and Translational Medicine, University of Tartu, 50411 Tartu, Estonia; kalle.kilk@ut.ee; 3School of Agriculture and Food Sustainability, Faculty of Science, The University of Queensland, Brisbane, QLD 4072, Australia; a.gambini@uq.edu.au; 4School of Veterinary Sciences, Faculty of Science, The University of Queensland, Brisbane, QLD 4072, Australia; 5Chair of Clinical Veterinary Medicine, Institute of Veterinary Medicine and Animal Sciences, Estonian University of Life Sciences, 51006 Tartu, Estonia; 6Chair of Veterinary Biomedicine and Food Hygiene, Institute of Veterinary Medicine and Animal Sciences, Estonian University of Life Sciences, 51006 Tartu, Estonia; esta.nahkur@emu.ee

**Keywords:** bovine embryo, metabolomics, in vitro, viability, developmental potential, biomarkers

## Abstract

Recent studies have shown that the metabolome of single embryo culture media is linked to successful pregnancy. In this study, the analysis was expanded to compare the metabolomes of viable and non-viable early-stage embryos and to examine metabolomic markers associated with hatching in viable embryos. The authors hypothesized that the metabolomic profiles of high-quality early blastocysts differ from those of non-viable embryos that reach the blastocyst stage but undergo developmental arrest at later stages. The metabolic profile of 43 spent bovine embryo culture medium samples were analyzed using liquid chromatography–mass spectrometry, covering 189 metabolites, including 40 acylcarnitines, 42 amino acids/biogenic amines, 91 phospholipids, 15 sphingolipids, and the sum of hexoses. Embryos were produced from abattoir-derived oocytes, and the culture medium samples were derived from Grade 1 early blastocysts that progressed to hatching (VBL; n = 10), non-viable early blastocysts that developed to the blastocyst stage but failed to hatch (DBL; n = 12), Grade 1 hatched blastocysts (HBL; n = 16), and plain growth media for control (CM; n = 5). It was observed that methionine sulfoxide (Met-SO) and lysophosphatidylcholine (lysoPC) C24:0 concentrations were significantly lower in the culture media from viable blastocysts compared to those from non-viable blastocysts (*p* < 0.001). Additionally, blastocysts that resulted in successful hatching had significantly lower levels of phospholipid, arginine (Arg), and methionine-related metabolites that significantly differentiated the control and viable blastocyst culture media from the media containing non-viable embryos. Building on previous studies, there appears to be an overlap in metabolites released during hatching that are also associated with successful pregnancy. The identified biomarkers can aid in assessing an embryo’s developmental potential and enhance embryo selection for transfer or cryopreservation.

## 1. Introduction

The in vitro production of bovine embryos has grown to the point where over a million embryos are transferred annually worldwide [1]. However, improving pregnancy outcomes following embryo transfer (ET) remains a challenge and depends on the non-invasive assessment of blastocysts [2]. Around 80% of the approximately 2 million bovine embryos produced in 2022 were created in vitro [3]. However, only about 40% of such transferred IVP embryos resulted in live births in 2020 [4]. The remaining 60% of pregnancy failures can be partly attributed to the transfer of poor-quality embryos, often resulting from inaccurate morphological evaluation using stereomicroscopy, which, currently, remains the primary method used by embryologists to assess the quality of in vitro-produced embryos [4,5]. This approach is subjective and does not fully evaluate the embryo based on its intrinsic qualities [6]. The current preference for in vitro embryo production in the cattle industry and the limitations of morphology-based embryo evaluation methods highlight the need for comprehensive studies of pre-implantation bovine embryos to identify viability and quality markers that can more accurately predict post-transfer embryo survival and pregnancy outcomes. The success of conception, proper embryo development, and the birth of healthy offspring are important for both dairy and beef farms [7].

Currently, three prominent research areas are gaining traction in the field of non-invasive embryo technology assessment: time-lapse microscopy enhanced by AI-based computational modeling [8]; the examination of extracellular vesicles (EVs) released into the culture media by blastocysts [9]; and, finally, the exploration of the correlation between metabolomic or proteomic markers in spent culture medium and embryo viability [10,11,12,13]. The metabolic shifts in the embryo’s early development reflect the cellular metabolic processes and overall health of the embryo [14]. Thus, the utilization of a metabolomic analysis of pre-implantation embryo growth media has demonstrated promising results in cattle and can provide several advantages [2,11,13]. These include reducing the risk of compromising the embryo and its environment, lowering the overall costs of in vitro production and embryo transfer by enhancing pregnancy rates, and reducing the need for open recipients. Recent research suggests that the metabolite profile of spent culture media may serve as a predictive indicator of pregnancy and birth outcomes in both cattle and humans [13,15,16]. It has been found that blastocysts leading to successful embryo implantation exhibited notably elevated levels of Met-SO, dihydroxyphenylalanine (DOPA), and spermidine and a low acetylcarnitine-to-free-carnitine ratio and C2 + C3-to-free-carnitine ratio while demonstrating decreased levels of threonine (Thr) and phosphatidylcholine PC ae C30:0 compared to control media. Conversely, in comparison to embryos that did not successfully implant, only DOPA, spermidine, C2/C0, (C2 + C3)/C0, and PC ae C30:0 levels exhibited significant differentiation [15].

It is well known that several factors can contribute to implantation failure after embryo transfer, regardless of embryo quality. These factors include the health status of the recipient animal, reduced endometrial receptivity, synchrony between the embryo and the uterus, early- and mid-luteal progesterone concentrations, the hormonal readiness of the recipient, errors in the embryo transfer technique, or a combination of these influences [5,17,18,19]. Taking these factors into account, the primary objective of this study was to assess the metabolomic profile of culture media from individual bovine embryos at a range of developmental stages and to investigate its correlation with developmental potential within a controlled laboratory setting, excluding embryo transfers to eliminate the influence of recipient animals. This included examining outcomes such as morphological blastocyst quality and the ability of embryos to hatch in vitro, a prerequisite for uterine attachment and the initiation of placentation. Metabolic signatures have the potential to predict embryo viability and facilitate the development of a non-invasive test to select single embryos for transfer or cryopreservation. The present study examined the efficacy of targeted metabolomic analysis by utilizing the AbsoluteIDQ^®^ p180 Targeted Metabolomics Kit in conjunction with liquid chromatography–tandem mass spectrometry (LC-MS/MS). This method enables the thorough collection of a wide array of molecular content within the culture media of individually cultured bovine embryos [15]. The authors hypothesized that the metabolomic profiles of high-quality early blastocysts differ from those of non-viable embryos that reach the blastocyst stage but experience developmental arrest at later stages.

## 2. Results

The full list of metabolites and their levels by study groups is given in Appendix A. The rates of cleavage, blastocyst formation, and hatching observed in our laboratory are presented in Appendix A.

### 2.1. Metabolites Differing in Culture Media of Viable and Non-Viable Blastocysts

In total, 2 metabolites showed highly significant differences (*p* < 0.001), and 29 exhibited significance (*p* between 0.05 and 0.001) between the culture media of the viable and non-viable blastocysts and empty controls (Table 1).

#### 2.1.1. Amino Acids and Derivative Metabolism

The concentrations of Met-SO and acetylornithine (Ac-Orn) in the culture media of degenerating blastocysts were significantly higher than those in the culture media of viable embryos. Asymmetric dimethylarginine (ADMA) and creatinine were undetectable in the empty media and the culture media of viable or hatched blastocysts but were present in the media of blastocysts that underwent developmental arrest. Kynurenine was detected in the media from non-viable blastocysts and in a small percentage of media from the hatched blastocysts. Viable early blastocysts, however, did not secrete any kynurenine. Citrulline (Cit) was not detectable in the empty media, nor in the culture media of viable blastocysts and hatched blastocysts, but was present in the media of blastocysts that underwent developmental arrest.

#### 2.1.2. Lipid Metabolism

Among acylcarnitines, the uptake of acetylcarnitine (C2), propionylcarnitine (C3), and valerylcarnitine (C5) was higher in viable blastocysts compared to degenerating blastocysts. Similarly, among glycerophospholipids, the uptake of lysophosphatidylcholine C24:0 (LysoPC a C24:0) and phosphatidylcholine C42:4 (PC aa C42:4) was significantly higher in viable blastocysts compared to blastocysts that experienced developmental arrest at a later stage.

### 2.2. Metabolite Sums and Ratios Differing Between Culture Media of Viable and Non-Viable Blastocysts

The summary concentration of metabolites that share a metabolic pathway or the ratios of the substrates and products of a pathway may be better indicators of metabolic activity than individual metabolites themselves [2]. The oxidized-methionine-to-methionine ratio (Met-SO/Met) indicated metabolic activity that significantly differentiated the control and viable blastocyst media from the media containing embryos that underwent developmental arrest (Table 2). The ratio of total hydroxylated sphingomyelins to non-hydroxylated sphingomyelins was significantly lower in the culture media of non-viable blastocysts compared to the culture media from viable embryos, where the ratio resembled that in the control (empty) media.

Elevated ratios of MUFA PC/SFA PC and PUFA PC/SFA PC were observed in the culture media of blastocysts that underwent developmental arrest, while decreases in the concentrations of PC ratios were noted in the culture media of viable and hatched blastocysts compared to the control media. The concentration of hexoses, predominantly glucose, was lower in the culture media from non-viable embryos.

### 2.3. Metabolites Differing Between the Culture Media of Viable Early and Hatched Blastocysts

Hatching is one step closer to successful implantation after the embryo survives the first week of development. Although the *zona pellucida*, which is shed during hatching, is unlikely to hinder metabolite trafficking, metabolic adjustments may still occur simultaneously. We observed an increase in the concentration of lysophosphatidylcholines (LysoPC a C24:0, LysoPC a C20:4), phosphatidylcholine (PC ae C40:2), putrescine, spermine, spermidine, and C3 and a decrease in the concentrations of Met-SO and LysoPC a C26:0 in the culture media of hatching blastocysts compared to viable early blastocysts.

## 3. Discussion

### 3.1. Lipid Metabolism

From the list of metabolites analyzed, various lipids were the most prominent in distinguishing between viable and non-viable or viable and hatching blastocysts. The diverse roles of lipids in early embryo development have been recently reviewed [20].

Acylcarnitines are intermediates in fatty acid beta oxidation and thus cellular and embryonic energetics [21,22]. The results of the current study suggest that an increase in the concentration of short-chain acylcarnitines, such as C2, C3, and C5, in the culture media indicates degenerating embryos. The carnitine esters of short-chain fatty acids can originate from the beta oxidation of longer fatty acid residues and also from amino acid catabolism, such as the breakdown of branched-chain amino acids (BCAAs) [23]. BCAAs play a vital role in embryo development, primarily in the synthesis of other amino acids [24]. Thus, an excessive concentration of C5 in embryo culture media can be a warning sign of errors in embryo protein synthesis.

A significant difference was observed in the levels of twelve PCs and in one lysoPC in the culture media of viable versus non-viable blastocysts. The concentrations of PCs, with one exception, were lower in the culture media of viable early blastocysts compared to the embryos that underwent developmental arrest. When comparing the individual PC species and sums and ratios, it seems that non-viable embryos release more PCs with unsaturated fatty acids. As fundamental structural elements of plasma lipoproteins and cell membranes, phosphatidylcholines play crucial roles in regulating cell function and signaling [24] Unsaturated fatty acid residues can be used for signaling, while saturated fatty acids are the energy reserve of the cells. In this case, the environment of blastocysts with developmental arrest is enriched with unsaturated fatty acid-containing PCs. The findings can be explained by either the increased consumption of saturated lipids for energy or the overproduction of monounsaturated lipids.

Blastocyst hatching on day 8 of development was associated with fewer changes in lipid composition. LysoPCs were more likely to appear here, implying more intense lipid catabolism during this period. Free fatty acids, most likely originating from acylglycerols, have been previously identified as potential markers of viability in the later stages of embryo development [13].

### 3.2. Monosaccharide Metabolism

The present study analyzed the concentration of the sum of hexoses in culture media containing viable early embryos and blastocysts that underwent developmental arrest. Interestingly, non-viable embryos consumed significant amounts of hexoses. The dominant hexose in the bovine culture media is glucose. During IVM, glucose metabolism, involving glycolysis and the pentose phosphate pathway (PPP), provides substrates essential for ooplasmic integrity and regulates oocyte meiotic maturation [25]. Glucose serves as a substrate for ATP and NADPH production once biosynthesis gears up after the initial embryo cleavages [25]. Although glucose is known to adversely affect early-stage embryo development, it serves as a critical energy substrate for embryos at the compacted morula and blastocyst stages. Studies in human pre-implantation embryos suggest that glucose may not be essential for development, but embryos that successfully develop to the blastocyst stage tend to exhibit higher glucose uptake [26]. In some studies, the level of glucose consumption has been linked to the quality and success of pregnancy [27]. The authors propose that the increased uptake of hexoses from culture media by degenerating embryos reflects the inability of mitochondria to balance ATP supply and demand. Further studies are needed to establish a causal association between the consumption or secretion of hexoses and embryo viability.

### 3.3. Amino Acid and Derivative Metabolism

Previous studies have demonstrated that the consumption of essential amino acids can vary among embryos. In a recent study on bovine pregnancy prediction [2], it was discovered that the concentrations of glutamine (Glu), proline (Pro), Met, Arg, lysine (Lys), and Thr in embryo culture media exhibit the highest pregnancy-predictive capability, albeit when assessing a single metabolite at a time. Lechniak et al. [11] compared various culture setups and observed significant differences in the metabolic pathways of phenylalanine (Phe), tyrosine (Tyr), Met, aspartate (Asp), Arg, Pro, and histidine (His) based on the culture system employed.

In our study, the majority of amino acids showed no significant differences between media containing viable embryos compared to empty control media. This suggests that the consumption and secretion of amino acids by embryos were relatively moderate relative to their levels in the growth media. Yet, three metabolites or metabolite groups related to amino acid metabolism emerged as elevated in non-viable blastocysts.

The first and most significant metabolite was oxidized Met or Met-SO. Met itself is an essential amino acid that has been extensively studied in recent blastocyst quality assessments and embryo implantation studies [28]; it is noteworthy that several studies have reported an increase in the concentration of Met in viable embryos [29], as well as in the uterine lumen of pregnant animals [30,31]. Met oxidation to Met-SO is considered to protect cells from oxidative damage, although a more complex bioregulatory role cannot be excluded [31,32]. The twist is that the levels of Met-SO in the culture media of viable blastocysts, which later progressed to hatching, were the lowest, whereas Met-SO levels rose significantly in the culture media of blastocysts that underwent developmental arrest at later stages. Hence, the antioxidative capacity of viable embryos is sufficient to lower oxidation in the environment, but oxidation returns at the time of hatching. Previous research has shown that moderate levels of oxidative stress, along with an increase in Met-SO or the Met-SO-to-Met ratio, may be beneficial for successful embryo implantation [15].

The second amino acid with a notable reduction in an environment of viable blastocysts is histidine. It too is an essential amino acid. The third notable differentiator between the culture media of viable and non-viable blastocysts, consistent with the aforementioned studies, is Arg metabolism. Even though Arg itself falls short in significance, an entire group of Arg-related metabolites has emerged. ADMA is methylated arginine regulating nitric oxide (NO) generation from Arg. NO is produced when the enzyme nitric oxide synthase (NOS) catalyzes the oxidation of L-arginine to L-citrulline [33]. It has been discovered that nitric oxide is associated with lower embryo quality and poorer pregnancy outcomes in in vitro embryo production settings [34]. Cit in the culture media can be derived from Arg via NOS or from ornithine (Orn) through the breakdown of proline or glutamine/glutamate [35]. The physiological functions of citrullination in the early embryo remain poorly defined, although the presence of Cit in the culture media has been associated with transcriptional regulation and the DNA damage response [36].

Cit and Orn together with Arg are intermediates in the most important nitrogen catabolic pathway, the urea cycle. Ac-Orn has been previously found in bovine pre-ovulatory follicular fluid [37]. Ac-Orn is an intermediate in the amino acid synthesis pathway and serves as a precursor to ornithine, which enters the urea cycle. Consequently, Ac-Orn is indirectly linked to the biochemical pathways that produce Arg and Pro [38]. Non-viable embryos secreted Ac-Orn into the culture media, while viable blastocysts either consumed it slightly or maintained its level at the same as that in empty culture media.

Creatinine and polyamines are metabolites synthesized from Arg. In contrast to Cit, Orn, or ADMA, which can be reversibly converted to Arg, these are derivatives of Arg that cannot be converted back [39]. Their upregulation suggests that they are either intentionally synthesized or that there is an excess of Arg that cannot be utilized in more meaningful ways. One can hypothesize that a higher concentration of Arg-related metabolites in the culture media may reflect the disrupted development of embryos, the dysregulation of polyamine and amino acid production, and a higher rate of embryonic death.

A previous study on bovine pregnancy prediction found that Thr levels were significantly lower in media from successfully implanted embryos compared to empty control media; however, no significant differences were observed between the successful and failed implantation groups [15]. In the current study, no significant differences in Thr levels were observed related to the viability of an early embryo.

### 3.4. Polyamine Metabolism

Polyamines are vital molecules with numerous roles in embryogenesis. They stabilize DNA and support its transcription; stabilize mRNA and assist in its translation for protein synthesis; promote cell growth, proliferation, and migration; maintain cell membrane stability; bind ATP; regulate ion channel functions; and mediate receptor–ligand interactions [40]. In the current study, differences in polyamine concentrations in the culture media of viable and non-viable embryos were not significant, although putrescine and spermidine were detected in a few non-viable samples. When comparing the culture media of early embryos to those of hatched embryos, we found a marked increase in the levels of spermidine, spermine, and putrescine as hatching approached, while these metabolites were conspicuously absent in the media of viable early blastocysts. Curiously, spermine appeared exclusively in the culture media of embryos at the hatching stage, underscoring its unique association with this developmental phase. This is consistent with a previous study, which found significantly higher spermidine levels in the culture media of embryos that resulted in pregnancies. Specifically, 90.91% of culture media from pregnancy-yielding embryos contained spermidine, compared to only 20% in non-yielding embryos (*p* = 0.00035) [15].

### 3.5. Summary

Figure 1 presents a diagram summarizing the key metabolomic pathways in various developmental stages and their associations with embryo viability. A high rate of oxidative stress markers, increased glucose consumption, elevated phospholipid synthesis, and abnormal Arg metabolism were observed in the culture media of early day 6 or 7 embryos that developed into blastocysts but failed to progress to hatching. Culture media from viable early blastocysts exhibited an increase in lipid catabolism. As hatching approached, a lower concentration of oxidative stress markers, an increased production of polyamines, and intensified lipid catabolism were observed.

### 3.6. Limitations and Future Research Directions

Despite various metabolites being proposed as embryo viability markers, available results remain limited, reproducibility is challenging, and their application across thousands of embryo production facilities is impractical. However, this study provides a comprehensive overview of potential metabolomic biomarkers for selecting viable early embryos from blastocysts undergoing developmental arrest and describes metabolic changes in the culture media of hatching blastocysts. Further research is needed to develop practical, user-friendly benchtop tests that leverage insights from metabolomic-based studies. To confirm the predictive value of biomarkers, the identified biomarkers should be tested in an independent dataset with similar or comparable conditions to the original one, as this study did not evaluate them in an external dataset. Additionally, it is important to acknowledge that this study lacks mechanistic validation exploring the roles of the discovered metabolites in cellular processes, embryo growth, viability, and differentiation. Our aim was to conduct a preliminary investigation to identify potential biomarkers, providing a solid foundation for future research. This should be pursued in future studies.

In the current study, oocytes for embryo production were obtained from abattoir ovaries, a factor known to influence oocyte quality and blastocyst rates, as indicated in previous studies [41,42]. Specifically, many crucial factors affecting oocyte and embryo quality—such as the donor’s age, the stage of the estrous cycle, nutritional status, genetic potential, and the presence of reproductive disorders—are often unknown. This aspect could be explored in future studies by running similar experiments using oocytes from live donors, without embryo transfers, to explore hatching outcomes and viable blastocyst development within a controlled laboratory environment, thereby eliminating the influence of the recipients.

Both human and bovine data suggest that the sex of embryos may influence metabolism [43,44]—an important consideration for future research. Metabolic differences between male and female bovine embryos may be reflected in the culture medium, potentially leading to the identification of metabolomic biomarkers for fast, precise, and non-invasive embryonic sex determination.

The total number of analyzed samples (n = 43) in this study was relatively small, primarily due to the size of the AbsoluteIDQ^®^ p180 Targeted Metabolomics Kit (biocrates life sciences ag, Innsbruc, Austria), which consists of a 96-well filter plate with integrated internal standards, calibration standards, and quality controls, allowing for a total of 80 samples to be analyzed. The small sample size may reduce the statistical power of this study; therefore, further research with a larger sample size is warranted.

## 4. Materials and Methods

### 4.1. Media

Serum-free media for all steps of in vitro embryo production, including in vitro maturation, fertilization, and the individual cultivation of embryos, were obtained from IVF Limited T/A IVF Bioscience (Bickland Industrial Park, Falmouth, Cornwall, UK). All media for the experiments were selected from the same production batch.

### 4.2. Experimental Design

Bovine embryos derived from slaughterhouse oocytes were used to compare the metabolic fingerprints of the culture media from viable early blastocysts, hatched blastocysts, and blastocysts that underwent developmental arrest at a later stage. Overall, six independent replicates of the IVF procedures were conducted, with each replicate comprising 100 zygotes. Culture medium samples were collected from individually cultured embryos on days 6, 7, and 8 of culture. The samples were derived from IETS Grade 1 early blastocysts that successfully progressed to hatching at a later stage (VBL; n = 10), IETS Grade 1 day 8 hatched blastocysts (HBL; n = 16), early blastocysts from day 6 or 7 that developed into blastocysts but failed to progress to subsequent hatching (DBL; n = 12), and plain culture media used as controls (CM; n = 5), resulting in a total of 43 samples.

### 4.3. Oocyte Collection and In Vitro Maturation

Ovaries were collected from Estonian Holstein cattle at a local slaughterhouse (HK Scan Estonia Inc., Rakvere, Estonia). Cumulus oocyte complexes (COCs) were aspirated from 2 to 7 mm follicles using an 18-gauge needle (B. Braun Melsungen AG, Hessen, Germany). COCs with three or more layers of unexpanded cumulus cells and a morphologically bright, evenly granulated cytoplasm were selected for in vitro maturation [41]. COCs were cultured in groups of fifty oocytes in 500 µL of BO-IVM medium, incubated at 38.5 °C with 5% CO_2_ in humidified air for 24 h.

### 4.4. In Vitro Fertilization and Cultivation

In vitro fertilization and cultivation were carried out according to our previous report [15], with the only modification being that zygotes were cultured for up to 8 days. In brief, frozen-thawed semen (commercially produced by the Animal Breeders’ Association of Estonia, Keava) from a bull, EHF ZIARD 27,481 (ID EE13993023), was used for IVF. The matured oocytes and sperm were co-incubated at a final concentration of approximately 1 × 10^6^ motile sperm/mL at 38.5 °C with 5% CO_2_ in air with maximum humidity. After 18 h, zygotes were denuded and cultured in 60 μL BO-IVC culture medium droplets (one zygote per droplet) overlaid with mineral oil in 90 mm Petri dishes (Sigma-Aldrich, St. Louis, MO, USA) at 38.5 °C under an atmosphere of 5% CO_2_, 5% O_2_, and 90% N_2_ for up to 8 days. Controls were created by placing 60 μL of plain BO-IVC culture medium droplets that were never in contact with an embryo under oil, and they were kept under the same conditions as zygotes. At 48 h post-IVF, droplets were examined to record embryo cleavage data. All uncleaved oocytes were discarded from this experiment.

### 4.5. Collection of Media for LC-MS/MS and Categorization of Samples

Culture medium samples (20 µL) were collected from droplets of individually cultured blastocysts at days 6, 7, and 8. Morphological assessments were conducted on each embryo at the time of medium collection, adapted from previously published work [45,46]. Any embryos arrested in development by day 6 were excluded from this experiment. Subsequently, samples were categorized according to their hatching outcomes: “VBL” refers to viable early blastocysts on day 6 or 7 that later progressed to hatching on day 8, “HBL” refers to hatched blastocysts, and “DBL” designates non-viable early blastocysts that developed to the blastocyst stage but failed to progress to hatching. Samples of plain culture media were collected on day 8 and treated as controls (hereafter referred to as “CM”), following the methodology established in previous research [47]. In summary, this indicates that for the group of viable embryos, culture medium samples were collected twice from the same embryos: once during the blastocyst stage on day 6 or 7 and again after successful hatching on day 8. In contrast, for the “DBL” group, culture medium samples were collected only once during the blastocyst stage, as these embryos developed into blastocysts but failed to hatch and were categorized as non-viable at the end of the culture period. To provide a more heterogeneous sample group, eliminate the influence of individual embryo variability, and account for the potential impact of embryo sex on metabolism, samples from individual embryos were not distinguished during categorization but were instead categorized solely based on developmental outcomes. All collected medium samples were labeled and stored immediately after collection at −20 °C (Figure 2).

### 4.6. Preparation of Culture Medium Samples for LC-MS/MS and Spectrometry

Each 20 µL culture medium sample underwent preparation according to the sample preparation protocol of The AbsoluteIDQ^®^ p180 Targeted Metabolomics Kit (biocrates life sciences ag, Innsbruc, Austria) for Agilent Infinity high-performance liquid chromatography (Agilent, Santa Clara, CA, USA) coupled to a 4500 QTRAP^®^ ion trap mass spectrometer (Sciex, Framingham, MA, USA). In brief, 10 µL of the sample, quality control, and calibration solution was dried on filter paper that had been imbued with isotope-labeled internal standards. The dried filter papers were then soaked with phenylisothiocyanate (Sigma-Aldrich, St. Louis, MO, USA) (3:19:19:19, water–ethanol–pyridine) for 1 h to derivatize the amino groups. After drying under a nitrogen flow, metabolites were extracted with 5 mM ammonium acetate (Sigma-Aldrich, St. Louis, MO, USA) in methanol for 30 min. Lipids were analyzed by direct infusion into mass spectrometry. For polar solvents, the extract was diluted with water (1:1), and a C18 column supplied by the kit’s manufacturer was used for metabolite separation.

### 4.7. Statistical Analysis

All statistical analyses were carried out using R version 4.2.0 (R Foundation for Statistical Computing, Vienna, Austria) [13]. In brief, all missing values were attributed to being below the detection limit and were replaced with zero. Univariate comparisons were performed for each metabolite or derived index (e.g., metabolite sums or ratios) across the different medium samples. For metabolites with more than 30% missing values, a chi-squared test was used to assess whether the metabolite was significantly more frequently detectable in any of the study groups. When fewer than 30% of values were missing, a Shapiro–Wilk test was applied to determine whether the data followed a normal distribution. Based on the results, either an ANOVA with Tukey’s honestly significant difference post hoc test or the Kruskal–Wallis test with Dunn’s post hoc test was used. The results were considered statistically highly significant at *p* < 0.001, and *p*-values between 0.05 and 0.001 were considered significant.

## 5. Conclusions

In conclusion, our study affirms the significance of glucose, Met, His, Arg, polyamine, and lipid metabolisms as pivotal processes in blastocyst pre-implantation development. Based on our findings and the existing literature, these pathways appear to be the most promising candidates for providing biomarkers to predict blastocyst viability and assess the embryo’s potential for successful implantation. The specific metabolite identified as a biomarker is influenced by experimental conditions and culture media. Although individual metabolites may differ across studies, the core processes likely remain consistent. Undoubtedly, our findings regarding Met-SO, LysoPC a C24:0, Cit, Ac.Orn, PC aa C42:4, ADMA, creatinine, C2, C3, C5, and kynurenine metabolism, and their association with the bovine embryo development potential, warrant careful consideration and further investigation.

## Figures and Tables

**Figure 1 ijms-26-02362-f001:**
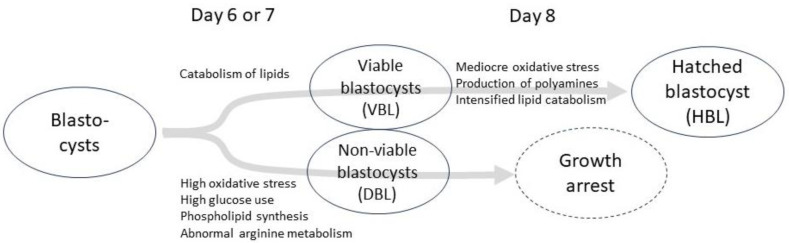
A diagram summarizing the main metabolomic pathways at different developmental stages and their associations with embryo viability. “DBL” refers to arrested early blastocysts that developed into blastocysts on day 6 or 7 but failed to progress to subsequent hatching. “VBL” denotes viable early blastocysts on day 6 or 7 that successfully progressed to hatching at a later stage. “HBL” represents viable day 8 embryos that had undergone hatching.

**Figure 2 ijms-26-02362-f002:**
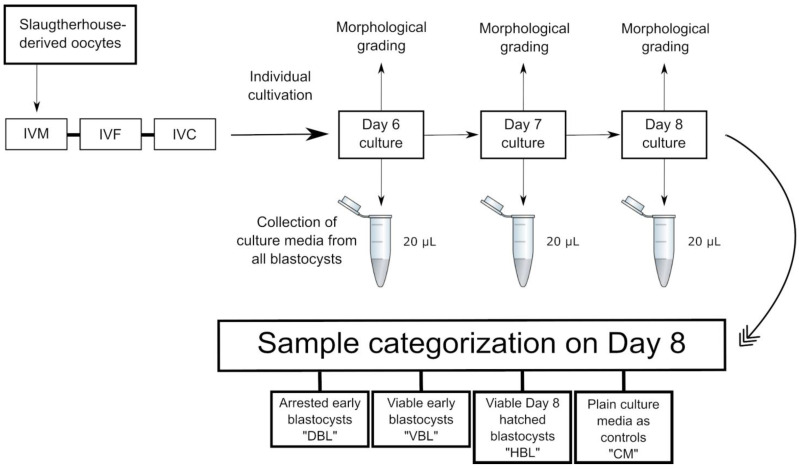
Sampling and categorization: “DBL” refers to culture media from arrested early blastocysts from day 6 or 7 embryos that developed into blastocysts but failed to progress to subsequent hatching. “VBL” denotes culture media from viable early blastocysts on day 6 or 7 that successfully progressed to hatching at a later stage. “HBL” represents culture media from viable day 8 embryos that had undergone hatching. Additionally, plain culture medium samples were collected on day 8 and processed as controls.

**Table 1 ijms-26-02362-t001:** Metabolites exhibiting significant differences in concentrations (highly significant differences are indicated by *p*-value in bold). In instances where metabolites were detected in only a limited number of samples, a chi-squared test was employed, and the count of samples with detectable concentrations was provided. The Kruskal–Wallis test, coupled with a Dunn post hoc test, was utilized if not indicated otherwise. “DBL” refers to culture media from arrested early blastocysts from day 6 or 7 embryos that developed into blastocysts but failed to progress to subsequent hatching. “VBL” denotes culture media from viable early blastocysts on day 6 or 7 that successfully progressed to hatching at a later stage. “HBL” represents culture media from viable day 8 embryos that had undergone hatching. “CM” refers to plain culture medium samples that were collected on day 8 and processed as controls.

Metabolites		Concentrations in Culture Media			
Metabolite	DBL	VBL	HBL	CM	*p*-Value
Met-SO (µM)	0.58 (0.45–4.68) ^a^	0.09 (0.059–0.144) ^b^	0.34 (0.17–0.53) ^a^	0.22 (0.14–0.62) ^ab^	**0.00012**
LysoPC a C24:0 (nM)	86 ± 32 ^a^	45 ± 13 ^b^	70 ± 20 ^a^	83 ± 5 ^a^	**0.00075 ^†^**
LysoPC a C20:4 (nM)	30 ± 11 ^ab^	18 ± 10 ^a^	36 ± 17 ^b^	15 ± 2 ^a^	0.0021 ^†^
Cit	5 (41.67%) ^a^	0 (0%) ^b^	0 (0%) ^b^	0 (0%) ^b^	0.0022 ^#^
PC.aa.C36.5 (nM)	3.2 (2–7) ^a^	1.7 (1.3–2.0) ^b^	1.3 (0.9–2.0) ^ab^	2.0 (1.0–2.0) ^a^	0.00242
Ac-Orn (µM)	0.18 (0.14–0.87) ^a^	0.12 (0.11–0.12) ^b^	0.13 (0.11–0.19) ^ab^	0.13 (0.12–0.19) ^ab^	0.00437
PC aa C42:4 (nM)	4 ± 2 ^a^	2 ± 1 ^b^	3 ± 1 ^ab^	2 ± 1 ^ab^	0.0062 ^†^
Putrescine	2 (16.67%) ^a^	0 (0%) ^a^	9 (56.25%) ^b^	3 (60%) ^a^	0.0073 ^#^
Spermine	0 (0 %) ^a^	0 (0 %) ^a^	6 (37.5 %) ^b^	0 (0 %) ^a^	0.0082 ^#^
Spermidine	4 (33.33 %) ^ab^	0 (0 %) ^a^	9 (56.25 %) ^b^	0 (0 %) ^a^	0.0085 ^#^
ADMA, Creatinine, Serotonin, t4-OH-Pro, Taurine	4 (33.33 %) ^a^	0 (0 %) ^b^	0 (0 %) ^b^	0 (0 %) ^b^	0.0098 ^#^
C2 (µM)	0.27 (±0.08) ^a^	0.195 (±0.024) ^b^	0.23 (±0.034) ^ab^	0.23 (±0.017) ^ab^	0.011 ^†^
C3 (nM)	49 (40–130) ^a^	41 (36–43) ^b^	50 (44–58) ^a^	54 (45–57) ^ab^	0.013
C5 (nM)	60 ± 23 ^a^	45 ± 7 ^b^	45 ± 6 ^b^	42 ± 2 ^ab^	0.015 ^†^
PC aa C34:1 (nM)	23 (15–96 ) ^a^	14 (13–15) ^ab^	17 (13–18) ^ab^	19 (15–21) ^ab^	0.015
PC aa C36:3 (nM)	7.2 (5.4–9.5) ^a^	4.7 (4.0–5.6) ^b^	5.0 (3.7–5.6) ^ab^	4.1 (4.0–5.2) ^a^	0.017
PC ae C42:0 (nM)	215 ± 12 ^ab^	226 ± 11 ^a^	211 ± 15 ^b^	223 ± 6 ^ab^	0.027 ^†^
PC ae C40:2 (nM)	2.7 (2.0–19.5) ^a^	1.8 (1.4–2.3) ^b^	2.0 (1.8–3.0) ^b^	3.0 (3.0–3.0) ^ab^	0.027
His (µM)	53.4 (51.3–55.7) ^a^	44.7 (40.6–49.0) ^b^	50.4 (45.2–54.4) ^ab^	49.5 (48.5–54.7) ^ab^	0.028
PC ae C32:2 (nM)	15 ± 8 ^a^	10 ± 3 ^b^	11 ± 1 ^ab^	10 ± 1 ^ab^	0.028 ^†^
LysoPC a C26:0 (nM)	23 ± 10 ^a^	26 ± 9 ^a^	17 ± 7 ^b^	18 ± 4 ^ab^	0.029 ^†^
PC aa C32:1 (nM)	9.0 (7.1–61) ^a^	5.9 (5.0–6.9) ^b^	6.7 (4.5–9.3) ^ab^	6.3 (6.0–6.4) ^ab^	0.029
PC aa C38:3 (nM)	4 (1–240) ^a^	1.1 (0.9–1.4) ^b^	1.7 (1.0–4.0) ^ab^	1.2 (1.0–1.4) ^ab^	0.0303
PC ae C38:2 (nM)	4.4 (3.0–24) ^a^	2.1 (1.5–2.8) ^b^	3.6 (2.7–5.3) ^ab^	4.0 (3.5–4.5) ^ab^	0.0374
PC ae C44:3 (nM)	10 (9–11) ^a^	13 (11–22) ^b^	13 (10–14) ^ab^	11 (10–17) ^ab^	0.0381
PC ae C38:1 (nM)	6.0 (3.8–27) ^a^	2.8 (2.4–3.4) ^b^	3.9 (3.0–5.0) ^ab^	3.0 (2.8–3.8) ^ab^	0.0423
PC aa C36:2 (nM)	58 (49–338) ^a^	48 (45–50) ^b^	50 (47–54) ^ab^	55 (54–57) ^ab^	0.0453

^a,b^ Culture medium groups that were not significantly different from each other are assigned the same letter, while groups with different letters indicate significant differences. ^†^ An ANOVA with Tukey’s HSD post hoc test was used as the test of comparison. ^#^ A chi-square test was used to compare frequencies.

**Table 2 ijms-26-02362-t002:** Metabolite sums and ratios exhibiting significant differences (highly significant differences are indicated by *p*-values in bold). In cases where metabolites were detected in only a few samples, a chi-squared test was employed, providing the count of samples with detectable concentrations. The Kruskal–Wallis test with a Dunn post hoc test was utilized if not indicated otherwise. “DBL” refers to culture media from arrested early blastocysts from day 6 or 7 embryos that developed into blastocysts but failed to progress to subsequent hatching. “VBL” denotes culture media from viable early blastocysts on day 6 or 7 that successfully progressed to hatching at a later stage. “HBL” represents culture media from viable day 8 embryos that had undergone hatching. “CM” refers to plain culture medium samples that were collected on day 8 and processed as controls.

Metabolites		Concentrations in Culture Media, μM			
Metabolite	DBL	VBL	HBL	CM	*p*-Value
Met-SO/Met	0.012 (0.009–0.09) ^a^	0.0016 (0.001–0.003) ^b^	0.006 (0.003–0.011) ^a^	0.004 (0.003–0.011) ^ab^	**6.3 × 10^−5^**
Total SM-OH/SM-non-OH	0.33 ± 0.16 ^a^	0.54 ± 0.11 ^b^	0.43 ± 0.13 ^ab^	0.57 ± 0.044 ^b^	**0.001** ^†^
C2/C0	0.14 (0.13–0.16) ^a^	0.14 (0.13–0.15) ^a^	0.16 (0.14–0.18) ^b^	0.16 (0.16–0.17) ^b^	0.00448
MUFA PC/SFA PC	0.37 (0.33–2.14) ^a^	0.32 (0.31–0.33) ^b^	0.32 (0.29–0.35) ^ab^	0.34 (0.31–0.38) ^a^	0.00602
Total PC ae	0.82 (0.80–1.48) ^ab^	0.82 (0.81–0.84) ^a^	0.79 (0.78–0.81) ^b^	0.81 (0.80–0.84) ^b^	0.0122
Hexoses	705 (670–830) ^a^	891 (838–946) ^b^	830 (777–879) ^ab^	816 (781–821) ^ab^	0.0221
SFA PC	0.80 (0.76–1.05) ^ab^	0.81 (0.79–0.82) ^a^	0.78(0.76–0.79) ^b^	0.80 (0.79–0.81) ^ab^	0.0313
PUFA PC/SFA PC	0.91 (0.86–2.44) ^a^	0.84 (0.82–0.86) ^b^	0.87 (0.84–0.91) ^a^	0.88 (0.84–0.89) ^a^	0.0389
Total AC-DC/Total AC	0.188 (±0.019) ^a^	0.202 (±0.006) ^b^	0.192 (±0.009) ^a^	0.198 (±0.006) ^a^	0.041 ^†^

^a,b^ Culture medium groups that were not significantly different from each other are assigned the same letter, while groups with different letters indicate significant differences. ^†^ An ANOVA with Tukey’s HSD post hoc test was used as the test of comparison.

## Data Availability

The data presented in this study are available on reasonable request from the corresponding author. The data are not publicly available due to privacy and/or ethical concerns.

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
