# Peer review of "Metabolomic Biomarkers in Bovine Embryo Culture Media and Their Relationship with the Developmental Potential of In Vitro-Produced Embryos"

_ijms, 2025, doi:10.3390/ijms26052362_

Round 1
Reviewer 1 Report (Previous Reviewer 1)
Comments and Suggestions for Authors
In present work, Tsopp et al. try to investigate the metabolomic biomarkers in bovine embryo culture media and their relationship to the developmental potential of in vitro produced embryos. This study demonstrated that methionine sulfoxide (Met-SO) and Lysophosphatidylcholine C24:0 concentrations were significantly lower in the culture media from viable blastocysts compared to those from non-viable blastocysts. In addition, blastocysts that resulted in successful hatching had significantly lower levels of phospholipid, arginine and methionine-related metabolites that significantly differentiated the control and viable blastocyst culture media from the media containing non-viable embryos. However, there are some questions that should be explained.
Major concerns
- At present, morphology-based embryo evaluation methods in vitro embryo production in the cattle is very fast and accurate. In addition, it is well known that the media of viable and non-viable blastocysts have significant differences. Therefore, in present study, it is inevitable that many identified biomarkers were found. In practice, the bovine in vitro embryo culture is not in a single embryo, there are many embryos (<50) in a liquid drop (200 ul). In fact, there are different grades (1, 2, 3, and 4) in a liquid drop after in vitro embryo culture, not only viable or non-viable blastocysts. In summary, we think that it is meaningless for analysing metabolites in culture media of viable and non-viable blastocysts.
- It is well known that embryo transfer is performed on day 6 or 7 at the stages of morula or early blastocyst. In this study, the hatched blastocysts at day 8 were used. The hatched blastocysts cannot be used for embryo transfer, and in vitro culture for hatched blastocyst is very difficult, because the in vivo hatched blastocyst will establish a connection with the uterus. Therefore, the experimental design for this study is unreasonable.
- Some ideas in this manuscript are not right. For example, ‘The remaining 60% of pregnancy failures can be partly attributed to the transfer of poor-quality embryos, often resulting from inaccurate morphological evaluation using stereomicroscopy’. Please provide some references.
- English grammar and writing style should be checked and revised throughout the manuscript.
Minor concerns
- In general, there should be no reference in the Abstract section (Line 24, ‘Tsopp E, et al. Metabolites, 2024’). In addition, writing style is not right.
- There should have a simple background for this study in the Abstract section.
- Line 42, ‘bovine’ is not suitable for a keyword.
- Lines 83-87, there are many factors can contribute to implantation failure, and immune reject by the mother should be included.
- Line 109, ‘(p < 0.001)’, ‘p’ should be used in italic. Please check these throughout this manuscript.
- Some reference format is not right. For example, ref. 13, no journal name. Some journal names are used in abbreviation, but some are not. Please check these throughout Reference section.
The English could be improved to more clearly express the research.
Author Response
Dear Reviewer,
Thank you for taking the time to review our manuscript and for providing valuable comments and suggestions.
Please find our responses below.
Major concerns
Comment 1: At present, morphology-based embryo evaluation methods in vitro embryo production in the cattle is very fast and accurate. In addition, it is well known that the media of viable and non-viable blastocysts have significant differences. Therefore, in present study, it is inevitable that many identified biomarkers were found. In practice, the bovine in vitro embryo culture is not in a single embryo, there are many embryos (<50) in a liquid drop (200 ul). In fact, there are different grades (1, 2, 3, and 4) in a liquid drop after in vitro embryo culture, not only viable or non-viable blastocysts. In summary, we think that it is meaningless for analysing metabolites in culture media of viable and non-viable blastocysts.
Response 1:
a) A considerable amount of research has been conducted on the inadequacy of morphological evaluation of embryos and its inability to accurately assess embryo viability. Morphological assessment provides only limited and insufficient information (Yaacobi-Artzi et al., 2024; Rabel et al., 2023). This approach is subjective and fails to evaluate embryos based on their intrinsic qualities. The high rate of pregnancy failures is partly due to the transfer of poor-quality embryos, a consequence of erroneous stereomicroscopy-based morphological evaluation. The increasing preference for in vitro embryo production in the cattle industry, combined with the limitations of morphology-based evaluation methods, underscores the need for comprehensive studies of pre-implantation bovine embryos. Identifying viability and quality markers that can more accurately predict post-transfer embryo survival and pregnancy outcomes was also the primary aim of this study.
Rabel, R.A.C.; Marchioretto, P.V.; Bangert, E.A.; Wilson, K.; Milner, D.J.; Wheeler, M.B. Pre-Implantation Bovine Embryo Evaluation—From Optics to Omics and Beyond. Animals 2023, 13, 2102. https://doi.org/10.3390/ani13132102
Yaacobi-Artzi, S.; Kalo, D.; Roth, Z. Morphokinetics of In Vitro-Derived Embryos—A Lesson from Human and Bovine Studies. Dairy 2024, 5, 419-435. https://doi.org/10.3390/dairy5030033
b) It would be incorrect to assume that we compared viable embryos to non-viable ones. I kindly encourage you to review the section "Collection of Media for LC-MS/MS and Categorization of Samples" more carefully. Morphological assessments were conducted on each embryo at the time of media collection. Any embryos arrested in development by day 6 were excluded from the experiment. Subsequently, samples were categorized according to their hatching outcomes: “VBL” refers to blastocysts on day 6 or 7 that later progressed to hatching on day 8, and “DBL” designates blastocysts that developed to the blastocyst stage on day 6 or 7, but failed to progress to hatching. This means that all samples were collected from morphologically viable blastocysts; however, many failed to develop further and were therefore categorized as non-viable. This, in turn, clearly highlights the insufficiency of morphological assessment in early embryos—not all embryos that reach the blastocyst stage are capable of further development.
c) Yes, some laboratories culture embryos in groups, primarily using slaughterhouse-derived material. However, in practice and commercial embryo production, all donors are cultured separately. Some are cultured in small groups, depending on the number of oocytes collected per donor, while others are cultured individually due to the use of time-lapse incubators or personal preference. Culturing oocytes and embryos individually is mandatory in most modern low-oxygen culture systems, where maturation and fertilization occur individually in microdroplets within a low-oxygen environment. Due to the high glucose consumption of oocytes and cumulus cells under low-oxygen conditions, group culturing is not feasible.
d) Regarding embryo grades, please refer to the "Experimental Design" section, where we have described this in detail. All samples were derived from IETS Grade 1 early blastocysts.
Comment 2: It is well known that embryo transfer is performed on day 6 or 7 at the stages of morula or early blastocyst. In this study, the hatched blastocysts at day 8 were used. The hatched blastocysts cannot be used for embryo transfer, and in vitro culture for hatched blastocyst is very difficult, because the in vivo hatched blastocyst will establish a connection with the uterus. Therefore, the experimental design for this study is unreasonable.
Response 2: We kindly encourage you to review our manuscript more carefully. None of the embryos were transferred since the in vitro embryo production was performed using abattoir-derived oocytes. It is well known that several factors can contribute to implantation failure after embryo transfer, regardless of embryo quality. These factors include the health status of the recipient animal, reduced endometrial receptivity, synchrony between the embryo and the uterus, early- and mid-luteal progesterone concentrations, the hormonal readiness of the recipient, errors in the embryo transfer technique, or a combination of these influences. Taking these factors into account, the primary objective of this study was to assess the metabolomic profile of culture media from individual bovine embryos at a range of developmental stages and to investigate its correlation with developmental potential within a controlled laboratory setting, excluding embryo transfers to eliminate the influence of recipient animals. That’s why we assessed the metabolomic profile of Day 6 and Day 7 embryos, and hatching was evaluated to confirm embryo viability in a laboratory setting.
A side note about the in vitro culture of hatched blastocysts: Eight day in vitro culture of bovine blastocysts is not difficult and is quite feasible, as the implantation process in bovine embryos begins around Day 16-18, after embryo elongation, with implantation starting on Day 19-20 and placentation starting after Day 21. Maternal recognition of pregnancy in cattle is on Day 15-16.
Comment 3: Some ideas in this manuscript are not right. For example, ‘The remaining 60% of pregnancy failures can be partly attributed to the transfer of poor-quality embryos, often resulting from inaccurate morphological evaluation using stereomicroscopy’. Please provide some references.
Response 3: Thank you for pointing this out. The references have been added: line 56.
Comment 4: English grammar and writing style should be checked and revised throughout the manuscript.
Response 4: The manuscript was proofread by our colleague, Prof. David Richard Arney (https://www.etis.ee/CV/David_Arney/eng/), a native English speaker and an expert in academic writing. Therefore, we believe that additional language editing services will not significantly improve the manuscript.
Minor concerns
Comment 1: In general, there should be no reference in the Abstract section (Line 24, ‘Tsopp E, et al. Metabolites, 2024’). In addition, writing style is not right.
Response 1: In our perspective, this article represents an extension of the previous work. Consequently, we propose including a reference to the earlier article within the abstract. The writing style has been corrected.
Comment 2:
Response 2: We have added lines 26-28.
Comment 3: Line 42, ‘bovine’ is not suitable for a keyword.
Response 3: "Bovine" has been changed to "Bovine embryo"
Comment 4: Lines 83-87, there are many factors can contribute to implantation failure, and immune reject by the mother should be included.
Response 4: The factors contributing to implantation failure are already discussed in lines 88-91, right after the section you are referring to.
Comment 5: Line 109, ‘(p < 0.001)’, ‘p’ should be used in italic. Please check these throughout this manuscript.
Response 5: 'p' is italicized throughout the manuscript.
Comment 6: Some reference format is not right. For example, ref. 13, no journal name. Some journal names are used in abbreviation, but some are not. Please check these throughout Reference section.
Response 6: Thank you; the references have been checked and formatted.
Reviewer 2 Report (New Reviewer)
Comments and Suggestions for Authors
The article by Tsopp and colleagues, titled “Metabolomic Biomarkers in Bovine Embryo Culture Media and Their Relationship to the Developmental Potential of In Vitro Produced Embryos,” explores a significant topic in reproductive biotechnology by identifying metabolomic biomarkers in bovine embryo culture media. The study contributes to improving embryo selection for implantation through LC-MS/MS-based metabolite profiling, clear categorization of embryos into distinct groups, robust statistical analyses, and practical applications for both cattle breeding and potential human fertility treatments.
However, several weaknesses should be addressed:
- The total number of samples (n=43) is relatively small, which may reduce the statistical power of the study. This limitation must be explicitly disclosed.
- The identified biomarkers should be tested in an independent dataset to confirm their predictive value. If not feasible, this limitation should be clearly acknowledged.
- While the study establishes correlations between metabolite levels and embryo viability, it lacks mechanistic validation (e.g., functional assays to assess the role of these metabolites in embryonic development).
- The manuscript briefly mentions the potential influence of embryo sex on metabolism. Including a preliminary sex-based analysis would provide stronger evidence and enhance the study’s conclusions.
- The study uses oocytes from slaughterhouse-derived ovaries, yet factors such as maternal health status and reproductive history may significantly affect the results. Strengthening the inclusion criteria and explicitly addressing these confounders in the discussion would improve the study’s rigor and validity.
- Lines 29-33 should use the same nomenclature as in the table legend (DBL, VBL, HBL) for clarity and consistency.
- "Cit" (Citrulline) should be consistently formatted.
This study provides valuable insights into bovine embryo metabolomics and has the potential to improve embryo selection strategies. Addressing these concerns will strengthen the manuscript's impact and contribution to the field.
Author Response
Dear Reviewer,
Thank you for taking the time to review our manuscript and for providing valuable comments and suggestions. The revisions have been marked in yellow for your convenience.
Comment 1: The total number of samples (n=43) is relatively small, which may reduce the statistical power of the study. This limitation must be explicitly disclosed.
Response 1: Thank you for bringing this to our attention. We acknowledge that our sample size is relatively small. To clarify this point, we have revised the "Limitations and Future Research Directions" section as follows (lines 354-359):
The total number of analyzed samples (n=43) in this study was relatively small, primarily due to the size of the AbsoluteIDQ® p180 Targeted Metabolomics Kit, which consists of a 96-well filter plate with integrated internal standards, calibration standards, and quality controls, allowing for a total of 80 samples to be analyzed. The small sample size may reduce the statistical power of the study; therefore, further research with a larger sample size is warranted.
Comment 2: The identified biomarkers should be tested in an independent dataset to confirm their predictive value. If not feasible, this limitation should be clearly acknowledged.
Response 2: Thank you, we are in full agreement with your perspective. The identified biomarkers were not tested in an independent dataset that has similar or comparable conditions to the original one. We have clarified this limitation to the readers in the "Limitations and Future Research Directions" sections as follows (lines 339-342):
To confirm the predictive value of biomarkers, the identified biomarkers should be tested in an independent dataset with similar or comparable conditions to the original one, as this study did not evaluate them in an external dataset.
Comment 3: While the study establishes correlations between metabolite levels and embryo viability, it lacks mechanistic validation (e.g., functional assays to assess the role of these metabolites in embryonic development).
Response 3: Thank you for bringing this to our attention. Indeed, based on the available data, we can only speculate about the roles of these metabolites in embryo development. We acknowledge that this study lacks mechanistic validation; however, validating the roles of the discovered metabolites in cellular processes, embryo growth, viability, and differentiation would require a much larger study. Our aim was to conduct a preliminary investigation to identify potential biomarkers, providing a solid foundation for future research. Additionally, in practical applications of embryo selection, markers remain valuable even if their specific role is unclear. Nevertheless, we plan to pursue the mechanistic aspects of markers once we have the necessary funding and a larger number of embryos to perform functional assays. To clarify these points for the readers, we have added a section to "Limitations and Future Research Directions" as follows (lines 342-348):
Additionally, it is important to acknowledge that this study lacks mechanistic validation exploring the roles of the discovered metabolites in cellular processes, embryo growth, viability, and differentiation. Our aim was to conduct a preliminary investigation to identify potential biomarkers, providing a solid foundation for future research. This should be pursued in future studies.
Comment 4: The manuscript briefly mentions the potential influence of embryo sex on metabolism. Including a preliminary sex-based analysis would provide stronger evidence and enhance the study’s conclusions.
Response 4: Yes, both human and bovine data show that the gender of embryos might affect metabolism. However, embryo sexing requires biopsying the embryo and removing a blastomere, which can affect its viability and metabolism. For this reason, we did not use it in this study. Additionally, we chose not to perform embryo sexing to maintain heterogeneous groups and better observe overall metabolic differences between viable and non-viable embryos. Nonetheless, it remains an interesting option for future studies.
Comment 5: The study uses oocytes from slaughterhouse-derived ovaries, yet factors such as maternal health status and reproductive history may significantly affect the results. Strengthening the inclusion criteria and explicitly addressing these confounders in the discussion would improve the study’s rigor and validity.
Response 5: Thank you, we agree with your perspective. We have already briefly addressed this in a Discussion under "Limitations and Future Research Directions" section, lines 349-356. Since in vitro embryo production was performed using abattoir-derived oocytes, drawing conclusions about maternal health and reproductive history was not possible, as the animals were not individually identified, and the donor history remains unknown to us. It is known that many crucial factors affect oocyte and embryo quality, such as the donor's age, stage of the estrous cycle, nutritional status, genetic potential, and presence of reproductive disorders. However, in this study, we could not speculate on the maternal influence on embryo development. On the other hand, the inclusion criteria used in this study were based on individual embryo viability, developmental potential, and morphology, as only culture media from IETS Grade 1 embryos were selected for further metabolomic analysis. The influence of maternal health on embryo quality and metabolism is an intriguing aspect that should be explored in future studies by conducting similar experiments using oocytes from live donors.
Comment 6: Lines 29-33 should use the same nomenclature as in the table legend (DBL, VBL, HBL) for clarity and consistency.
Response 6: Thank you for noticing this. Lines 33-35 (previously 29-33) have been corrected accordingly.
Comment 7: "Cit" (Citrulline) should be consistently formatted.
Response 7: Thank you, this has been corrected.
Round 2
Reviewer 1 Report (Previous Reviewer 1)
Comments and Suggestions for Authors
Thanks for author’s responses. However, we do not agree with the responses of the authors, and do not think again that this manuscript is suitable for published in this journal.
- Our Comment 1: At present, morphology-based embryo evaluation methods in vitro embryo production in the cattle is very fast and accurate.
Authors Response is ‘A considerable amount of research has been conducted on the inadequacy of morphological evaluation of embryos and its inability to accurately assess embryo viability. Morphological assessment provides only limited and insufficient information (Yaacobi-Artzi et al., 2024; Rabel et al., 2023).’
Rabel, R.A.C.; Marchioretto, P.V.; Bangert, E.A.; Wilson, K.; Milner, D.J.; Wheeler, M.B. Pre-Implantation Bovine Embryo Evaluation—From Optics to Omics and Beyond. Animals 2023, 13, 2102. https://doi.org/10.3390/ani13132102
Yaacobi-Artzi, S.; Kalo, D.; Roth, Z. Morphokinetics of In Vitro-Derived Embryos—A Lesson from Human and Bovine Studies. Dairy 2024, 5, 419-435. https://doi.org/10.3390/dairy5030033
This reviewer requires that authors provide the reference papers from the high quality and very professional Journals (for example, BOR, Reproduction, or Theriogenology).
There is a recent paper from high quality and very professional Journal (Nature Medicine), which still focuses on morphology-based embryo selection in IVF.
Illingworth PJ, Venetis C, Gardner DK, Nelson SM, Berntsen J, Larman MG, Agresta F, Ahitan S, Ahlström A, Cattrall F, Cooke S, Demmers K, Gabrielsen A, Hindkjær J, Kelley RL, Knight C, Lee L, Lahoud R, Mangat M, Park H, Price A, Trew G, Troest B, Vincent A, Wennerström S, Zujovic L, Hardarson T. Deep learning versus manual morphology-based embryo selection in IVF: a randomized, double-blind noninferiority trial. Nat Med. 2024;30(11):3114-3120. doi: 10.1038/s41591-024-03166-5.
- Our Comment 1: In practice, the bovine in vitro embryo culture is not in a single embryo, there are many embryos (<50) in a liquid drop (200 ul). In fact, there are different grades (1, 2, 3, and 4) in a liquid drop after in vitro embryo culture, not only viable or non-viable blastocysts. In summary, we think that it is meaningless for analysing metabolites in culture media of viable and non-viable blastocysts.
Authors Response is ‘Yes, some laboratories culture embryos in groups, primarily using slaughterhouse-derived material. However, in practice and commercial embryo production, all donors are cultured separately. Some are cultured in small groups, depending on the number of oocytes collected per donor, while others are cultured individually due to the use of time-lapse incubators or personal preference. Culturing oocytes and embryos individually is mandatory in most modern low-oxygen culture systems, where maturation and fertilization occur individually in microdroplets within a low-oxygen environment. Due to the high glucose consumption of oocytes and cumulus cells under low-oxygen conditions, group culturing is not feasible.’
Authors response that ‘Culturing oocytes and embryos individually is mandatory’. However, in this study (Abstract section), ‘Grade 1 early blastocysts that progressed to hatching (n = 10), non-viable early blastocysts that developed to the blastocyst stage but failed to hatch (n = 12), Grade 1 hatched blastocysts (n = 16), and plain growth media for control (n = 5)’.
Line 390, ‘COCs were cultured in groups of fifty oocytes’, please explain.
In addition, the data of this study are from cultured in groups of blastocysts (n = 5, 10, or 12). Therefore, it is doubt if the conclusion can be used for single bovine embryo growth. The concentrations of metabolomic biomarkers will be very lower from the single bovine embryo than that in this study (n = 5, 10, or 12). Therefore, the conclusion of this study can not be used for evaluating the single bovine embryo culture.
- Our Comment 2: It is well known that embryo transfer is performed on day 6 or 7 at the stages of morula or early blastocyst. In this study, the hatched blastocysts at day 8 were used. Therefore, the experimental design for this study is unreasonable.
Authors Response is ‘Eight day in vitro culture of bovine blastocysts is not difficult and is quite feasible.’, ‘these blastocysts are not used for embryo transfer’.
It is well known that owing to the different environments between in vitro culture and in vivo culture, there is an increasing death rate with the increase in day in vitro culture of embryo. The aim of this paper is that some metabolomic biomarkers in vitro culture of bovine embryo is found to classify the good and bad embryos, in order to be used in embryo transfer on day 6 or 7. Therefore, in this study, the hatched blastocysts at day 8 were used, which is unreasonable.
Comments on the Quality of English LanguageThe English could be improved to more clearly express the research.
Author Response
Dear Reviewer,
Thank you for your time and feedback. We understand that some aspects of our manuscript may be complex and require a thorough understanding of both bovine embryology and the English language. We sincerely appreciate your effort in reviewing our work. We will carefully address your comments once again—please find our responses below.
Comment 1: This reviewer requires that authors provide the reference papers from the high quality and very professional Journals (for example, BOR, Reproduction, or Theriogenology). There is a recent paper from high quality and very professional Journal (Nature Medicine), which still focuses on morphology-based embryo selection in IVF.
Illingworth PJ, Venetis C, Gardner DK, Nelson SM, Berntsen J, Larman MG, Agresta F, Ahitan S, Ahlström A, Cattrall F, Cooke S, Demmers K, Gabrielsen A, Hindkjær J, Kelley RL, Knight C, Lee L, Lahoud R, Mangat M, Park H, Price A, Trew G, Troest B, Vincent A, Wennerström S, Zujovic L, Hardarson T. Deep learning versus manual morphology-based embryo selection in IVF: a randomized, double-blind noninferiority trial. Nat Med. 2024;30(11):3114-3120. doi: 10.1038/s41591-024-03166-5.
Response 1: Thank you, we have added the Nature Medicine article by Illingworth et al as a reference to support the claim, that morphology-based selection needs improvements. Illingworth et al say :
„ Currently, during the culture period, the embryologist removes the embryos from the incubator, usually more than once, and performs a ‘snapshot’ morphological assessment under a microscope3. This approach is time consuming, subjective and has, in essence, changed little since the first IVF birth 45 years ago4,5“
They aimed to reduce subjectivity of the method by replacing the human factor with AI. Their results are interesting, but hardly a breakthrough. We use an alternative quantitative and therefore objective analytical platform, i.e. metabolomics, to solve, essentially, the same problem.
The same concern is shared by most articles published in the IVF field regardless of journal impact factors. Limited scope dedicated journals often have more relevant expert articles than the wide-coverage high impact science journals meant for a general audience. We would like not to extend the reference list needlessly, and believe the current references are sufficient enough in both senses: relevance and impact.
Comment 2: Line 390, ‘COCs were cultured in groups of fifty oocytes’, please explain.
Response 2: In the Materials and Methods section, it is described that COCs (cumulus-oocyte complexes) were cultured together in groups of 50 oocytes per 500 µL of media during IVM. Similarly, IVF was performed in the group of oocytes. However, please pay attention that subsequently the presumptive zygotes were cultured individually. As described in Lines 400-402: „After 18 h, zygotes were denuded and cultured in 60 μL BO-IVC culture medium droplets (one zygote per droplet) overlaid with mineral oil in 90 mm Petri dishes...“ This approach allowed us to distinguish between different embryos and collect their individual culture media samples for metabolomic analysis starting from Day 6. This approach is commonly used in bovine in vitro embryo production, where oocytes are matured and fertilized in groups (to improve maturation and fertilization rates and reduce the amount of semen used for IVF), but embryo culture is carried out separately.
Comment 2.1: In addition, the data of this study are from cultured in groups of blastocysts (n = 5, 10, or 12). Therefore, it is doubt if the conclusion can be used for single bovine embryo growth. The concentrations of metabolomic biomarkers will be very lower from the single bovine embryo than that in this study (n = 5, 10, or 12). Therefore, the conclusion of this study can not be used for evaluating the single bovine embryo culture.
Response 2.1: There seems to be a misunderstanding. The embryos were not cultured in groups but individually, as we have clearly stated in our manuscript (lines 91–95, 378–379, 410–411). The numbers n = 5, 10, or 12 refer to the number of samples in each category.
For example, there were 5 + 10 + 12 = 27 growth media droplets. In the control group, five of these droplets were empty, containing no embryo. The remaining 22 droplets each contained a single embryo; of these, 10 developed to the hatching stage by day 8, while 12 failed to hatch by day 8.
Comment 3: Our Comment 2: It is well known that embryo transfer is performed on day 6 or 7 at the stages of morula or early blastocyst. In this study, the hatched blastocysts at day 8 were used. Therefore, the experimental design for this study is unreasonable.
Authors Response is ‘Eight day in vitro culture of bovine blastocysts is not difficult and is quite feasible.’, ‘these blastocysts are not used for embryo transfer’.
It is well known that owing to the different environments between in vitro culture and in vivo culture, there is an increasing death rate with the increase in day in vitro culture of embryo. The aim of this paper is that some metabolomic biomarkers in vitro culture of bovine embryo is found to classify the good and bad embryos, in order to be used in embryo transfer on day 6 or 7. Therefore, in this study, the hatched blastocysts at day 8 were used, which is unreasonable.
Response 3: As we already explained in our previous response, none of the embryos were transferred. In our opinion, this is clearly explained in the Introduction, Materials and Methods, and Future Research Directions sections of our manuscript.
The ability of embryos to hatch in vitro was used solely to examine developmental outcomes in a laboratory setting as successful hatching indicates developmental capacity of an embryo. This is explained in the manuscript on lines 91–97.
The primary objective of this study was to assess the metabolomic profile of culture media from individual bovine embryos at various developmental stages and to investigate its correlation with developmental potential within a controlled laboratory setting, explicitly excluding embryo transfers to eliminate the influence of recipient animals. This is why we assessed the metabolomic profile of Day 6 and Day 7 embryos, with hatching evaluated to confirm embryo viability in a laboratory setting. Additionally, we assessed metabolomic differences and requirements between early blastocysts and hatched blastocysts, which, to the best of our knowledge, has not been explored before.
Comments on the Quality of English Language: The English could be improved to more clearly express the research. Response: : The manuscript was proofread by our colleague, Prof. David Richard Arney (https://www.etis.ee/CV/David_Arney/eng/), a native English speaker and an expert in academic writing. Therefore, we believe that additional language editing services will not significantly improve the manuscript.
Reviewer 2 Report (New Reviewer)
Comments and Suggestions for Authors
The revision process has significantly enhanced the scientific quality of the manuscript, improving its clarity and rigor.
Author Response
Dear Reviewer,
We sincerely appreciate your valuable comments and feedback, which have helped us improve the manuscript. Thank you!
This manuscript is a resubmission of an earlier submission. The following is a list of the peer review reports and author responses from that submission.
Round 1
Reviewer 1 Report
Comments and Suggestions for Authors
In present work, Tsopp et al. try to investigate the metabolomic biomarkers in bovine embryo culture media and their relationship to the developmental potential of in vitro produced embryos. This study demonstrated that methionine sulfoxide (Met-SO) and Lysophosphatidylcholine C24:0 concentrations were significantly lower in the culture media from viable blastocysts compared to those from non-viable blastocysts. In addition, blastocysts that resulted in successful hatching had significantly lower levels of phospholipid, arginine and methionine-related metabolites that significantly differentiated the control and viable blastocyst culture media from the media containing non-viable embryos. However, there are some questions that should be explained.
Major concerns
1. This study only analysed metabolites in culture media of viable and non-viable blastocysts, metabolite sums and ratios in culture media of viable and non-viable blastocysts, and metabolites in the culture media of viable early and hatched blastocysts. However, the confirmation by embryo transfer is needed. Only metabolomics data are not enough to be published in this journal.
2. This study only provided the data of metabolites in the supplemental Table S1. However, many figures related to metabolite analysis should be provided.
3. In experimental design, the genetic basis and gender of blastocysts need to be considered.
4. English grammar and writing style should be checked and revised throughout the manuscript.
Minor concerns
1. As a scientific paper, it should be written in the third person. There are so many ‘we’. Please check these throughout this manuscript.
2. In general, there should be no reference in the Abstract section.
3. Lines 103-104, ‘p’ should be used in italic. Please check these throughout this manuscript.
4. Lines 334, ‘CO2’, subscript should be used. Please check these throughout this manuscript, including ‘O2’ and ‘N2’.
5. Page 10, a title for Figure 1 is needed.
6. The reference format is not consistent. Some references have doi, but some references are not. Please check these throughout Reference section.
Comments on the Quality of English LanguageThe English could be improved to more clearly express the research.
Reviewer 2 Report
Comments and Suggestions for Authors
General comments
The manuscript entitled “Metabolomic biomarkers in bovine embryo culture media and their relationship to the developmental potential of in vitro produced Embryos” by Tsopp et al. investigated the metabolic profiles of bovine embryo culture media samples using liquid chromatography-mass spectrometry. The study explored the differences in metabolic profile between non-viable early blastocysts and viable blastocysts, as well as between viable blastocysts and hatched blastocysts. As one technique of non-invasive embryo assessment, evaluation of spent medium provides valuable information about the embryo development stages. However, there are some drawbacks in the experimental design of this study. As the medium samples collected at day 6, 7, 8, it was unclear whether the no. of samples at each stage were collected from same embryos or different embryos. The authors could consider comparing the metabolic profile in individual embryos throughout the 3 days, using the metabolic profile at day 6 as the baseline. For data analysis, authors should consider the data collected from the same embryos as a factor in the statistical model, maybe repeated ANOVA is a better model. Moreover, no information about the replicates of IVF experiment. As shown in Table S2, there were 82 blastocysts, however, the number of spent medium samples (10-16) were much less than the number of the blastocysts. Please explain the differences in the numbers.
Specific comments:
Introduction, please add literature review about spent medium evaluation in other reports.
Table 1, please define DBL, VBL, HBL, CM. It’s better to put all the metabolites as category, such as lipid metabolism, monosaccharide metabolism, amino acid and derivative metabolism, which would be easier to follow the results.
L323-325, please describe which samples (DBL, VBL, HBL) were collected from which day (day 6, 7 or 8).
L352-353, after 20ul media collected from each drop, was the new medium added into the drop? If not, would less volume of the medium per drop affect the LC-MS/MS and Spectrometry results?
L364-368, please provide more details about the methods for sample preparation and how the samples was analyzed by LC-MS/MS and Spectrometry, which readers can follow the procedure.
Conclusion, rewrite the conclusion after the data reanalyzed.
Round 2
Reviewer 1 Report
Comments and Suggestions for Authors
Thanks for author’s responses. However, we do not agree to the responses of the authors.
1. This study only analysed metabolites in culture media of viable and non-viable blastocysts, metabolite sums and ratios in culture media of viable and non-viable blastocysts, and metabolites in the culture media of viable early and hatched blastocysts. However, the confirmation by embryo transfer is needed.
The author replied that ‘the long-term outcomes have already been published’. Therefore, we do not think again that this manuscript is suitable for published in this journal.
2. In the first review, we think that many figures related to metabolite analysis should be provided. However, the authors do not provide more figures. The reply was ‘a Venn diagram summarizing the metabolites that differentiate among the groups, which would result in a repetition of the results already presented in Tables 1 and 2’. Therefore, the data of this manuscript is very limited, and it is suitable for published in this journal.
3. Authors claimed that this manuscript was proofread by their colleague, Prof. David Richard Arney. However, there are still some low wrongs. For example, Line 381, ‘8 days[13]’. In addition, the writing style of all Table legends is not right. In general, there should be no reference in the Abstract section. However, there is still a reference in the Abstract section.
4. As a Communication paper, the percent match is 35%, which is very higher.
Comments on the Quality of English LanguageThe English could be improved to more clearly express the research.
Reviewer 2 Report
Comments and Suggestions for Authors
Most of my comments and concerns have been addressed. However, I have a couple more questions/suggestions.
1. As mentioned in the experimental design, VBL, n=10, but HBL, n=16. Does this indicate some of the medium samples were only collect at hatched stage (day 8), not at early stage (day 6 or 7)? Please explain the reason.
2. Figure 2 is confused, as not all the media samples were collected at both day 6 and 7, or day 8. Please adjust the figure according to experimental design.